# Clinical characteristics and the risk of hospitalization of patients with coronavirus disease 2019 quarantined in a designated hotel in Japan

**Hiromasa Ito**[1], **Tadafumi Sugimoto**[1☯], **Yoshito Ogihara**[1☯], **Tairo Kurita**[1☯], **Masaki Tanabe**[2☯], **Masahiro Hirayama**[3☯], **Shuji Isaji**[4☯], **Kaoru Dohi**[1] *

1 Department of Cardiology and Nephrology, Mie University Graduate School of Medicine, Tsu, Japan, 2 Department of Infection Control and Prevention, Mie University Hospital, Tsu, Japan, 3 Department of Pediatrics, Mie University Graduate School of Medicine, Tsu, Japan, 4 Director of Mie University Hospital, Tsu, Japan

☯ These authors contributed equally to this work.
* dohik@med.mie-u.ac.jp

**Data Availability Statement:** Data cannot be shared publicly because of the Mie Prefectural Government policy. Qualified researchers may

## Abstract

The aim was to investigate the clinical characteristics of coronavirus disease 2019 (COVID-19) patients who were admitted to a designated hotel, and to clarify the risk factors for hospitalization of such patients with clinical deterioration. The medical records of COVID-19 patients who were admitted to the designated hotel in Mie Prefecture, Japan, between August 2020 and September 2021 were reviewed retrospectively. Of the 1,087 COVID-19 patients who were admitted to the designated hotel, 936 patients (32.1± 12.8 years, 61.3% male) were recruited after excluding patients under the age of 15 years (n = 33), those admitted from the hospital (n = 111, 10.2%), COVID-19 vaccinated patients (n = 4, 0.4%), and those who were discharged to their own home due to social disorders (n = 3). During the study period, 884 patients (94.4%) were discharged to their own home with improving symptoms, whereas 52 patients (5.6%) were hospitalized for a deteriorating clinical condition. The logistic regression analyses showed that older age ($\geq$ 40 years), higher body mass index ($\geq$ 25 kg/m$^2$), hypertension were the risk factors for hospitalization. As the new risk scale score based on the results of the odds ratios increased, the hospitalization rate increased significantly: 2.0% at 0–1 points, 9.7% at 2–3 points, and 28.8% at 4–5 points (p < 0.001). None of the 52 hospitalized patients died, and none developed serious complications from COVID-19 after hospitalization. In conclusion, the designated accommodation program for COVID-19 patients was safe, especially for those with a low risk for hospitalization.

## Introduction

The coronavirus disease 2019 (COVID-19) pandemic, caused by severe acute respiratory syndrome coronavirus 2 (SARS-CoV-2), has become a global health and medical resources crisis

request access to this data from the Clinical Research Ethics Review Committee of Mie University Hospital (kk-sien@med.mie-u.ac.jp).

**Funding:** The authors received no specific funding for this work.

**Competing interests:** The authors have declared that no competing interests exist.

[1]. The first case of COVID-19 was reported in Japan on January 16, 2020. On February 2020, an outbreak of COVID-19 was identified on an international cruise ship that left Yokohama Port [2]. In this outbreak, 712 passengers and crew were positive on SARS-CoV-2 testing, and some positives were quarantined at a tourist hotel in Chiba. This hotel also accepted the temporary returnees from Wuhan, China since January 29, 2020.

On February 1, 2020, the Government of Japan classified COVID-19 as a designated infectious disease under the Infectious Diseases Control Law [3], and individuals with a positive test result for SARS-CoV-2 had to be quarantined in an isolation ward in a hospital based on the law. However, it became difficult to admit all positive patients to the hospital due to the rapid increase in new COVID-19 cases. Accordingly, the Ministry of Health, Labour and Welfare proceeded to prepare an accommodation program in designated hotels for their recovery, and the government's Novel Coronavirus Response Headquarters notified each Prefecture of the policy and operation manuals for hotel accommodation and home recuperation on April 2, 2020. In response to the notification, the designated accommodation program in a tourist hotel began from August 13, 2020, and use of a second designated hotel was started from June 15, 2021 in Mie Prefecture.

Therefore, the purpose of the present study was to investigate the clinical characteristics of COVID-19 patients who were admitted to a designated hotel, and to clarify the safety of the designated hotel accommodation program and identify the risk factors for hospitalization during outbreaks or pandemic periods using the medical records in Mie Prefecture, Japan.

## Materials and methods

### Study design and study population

This was a retrospective, single-cohort study that enrolled COVID-19 patients in the designated hotel in Mie Prefecture, Japan, between August 2020 and September 2021.

### The accommodation program at a designated hotel

The designated hotel accommodation programs were operated at two tourist hotels for COVID-19 patients in Mie Prefecture. In this study, one hotel, in which the designated hotel accommodation program was first operated, was analyzed. The designated hotel accommodation served as a temporary isolation and monitoring program until the patients had recovered from COVID-19. In principle, the coordinating team of the prefectural office has the authority to suggest whether patients can recuperate at home or should be admitted to the designated hotel based on the patient's age, general condition, and coexisting disorders reported from public health centers. Doctors and nurses were dispatched from the cooperating medical institutions, including Mie University Hospital, and they were stationed at the hotel to monitor patients' conditions and to respond to sudden deterioration. If the stationed doctors or nurses determined that a patient's symptoms or condition had worsened, the coordinating team of the prefectural office quickly arranged for hospital treatment (S1 Fig).

### Hotel admission criteria

Patients were allowed to be admitted to the hotel based on the following criteria in the 2nd and 3rd waves: under 40 years of age, asymptomatic or mild/improving symptoms, no pre-existing comorbidities (respiratory disorder, cardiovascular disease, hypertension, diabetes mellitus, and renal disorder), body mass index (BMI) $\leq 30$ kg/m$^2$, no immunodeficiency, no pregnancy, no moderate/severe pneumonia, and no specific abnormalities (renal failure, hepatic disorder, severe anemia, or coagulation disorder) other than an inflammatory reaction. Hotel admission

was permitted if the physician decided that the blood and radiographic examinations were not needed. Indeed, almost all patients (96.5%) who were admitted from their home did not undergo blood or radiographic examinations before the hotel admission.

The admission criteria were revised with every wave of COVID-19 infection (S1 Table); that is, the upper age limit for acceptance was raised to 50 years in the 4th wave and 65 years in the 5th wave. In addition, the admission criteria for the clinical condition including patient symptoms and comorbidities for acceptance were also relaxed in the 4th and 5th waves. The patients gave written, informed consent before hotel admission and were required to follow predetermined rules at the designated hotel [4].

## Definitions of patients' characteristics

Hypertension was diagnosed if peripheral blood pressure was > 140/90 mmHg or if the patient was prescribed medication for hypertension. Diabetes mellitus was diagnosed if hemoglobin A1c was > 6.5% or was assumed if the patient was prescribed medication for the treatment of diabetes mellitus. Heart disease was defined as heart failure, angina pectoris, or a history of myocardial infarction. Heart failure was diagnosed if the patient had a history of a hospitalization for heart failure, symptoms due to heart failure (New York Heart Association functional class $\geq 2$), or left ventricular ejection fraction < 40%.

Respiratory disease was defined as persistent lung disorders such as asthma, chronic obstructive pulmonary disease, or restrictive lung diseases. The severity of COVID-19 at admission was classified as mild, moderate, or severe. Patients with mild COVID-19 were defined as those who did not require oxygen, patients with moderate COVID-19 were defined as those who required oxygen, and patients with severe COVID-19 were defined as those who required intubation or extracorporeal membrane oxygenation [5].

## Recovery protocol at the hotel

The patients' vital signs including blood pressure, pulse rate, percutaneous oxygen saturation ($SpO_2$), and temperature were checked by the nurses when the patients arrived at the hotel. They were then moved to individual rooms after taking the short orientation and agreed to stay [6]. All patients were instructed to measure their temperatures and $SpO_2$ three times a day (morning, noon, and evening), and to report the results by telephone or the Health Center Real-time Information-sharing System (HER-SYS) [7]. Nurses were stationed around the clock in two shifts to observe the patients' health condition by telephone twice a day, morning and evening. The designated hotel is a non-medical facility, and, therefore, face-to-face medical care, including oxygen administration and intravenous fluid therapy, could not be provided. However, if a patient described symptoms or requested a medical consultation, the on-call doctor examined the patient by telephone or video. Thus, telemedicine, including drug prescription (without anti-viral drugs), could be provided. Tests for active SARS-CoV-2 infection were not conducted at the hotel. The patients could be discharged from the hotel 10 days after the onset of symptoms and 48 hours after the resolution of fever or respiratory symptoms had improved.

## Emergency protocol

If a patient developed cardiopulmonary arrest, shock, severe oxygen desaturation ($SpO_2 < 90\%$), or consciousness disturbance, the doctor or nurse immediately came to the patient's room and called for transport of the patient to the emergency hospital.

The doctor or nurse also arranged for transportation of a patient to the hospital when the coordinating team of the prefectural office recommended it, if the patient developed sudden

onset dyspnea, oxygen desaturation (SpO$_2$ < 93%), chest pain, orthopnea, wheezing, difficulty breathing, ill complexion, persistent fever ($\geq$ 38.5°C more than 2 consecutive days), inadequate oral intake (less than 30% for more than 2 days), difficulty walking, or wound/fractures that required treatment. All patients who met the above criteria were transferred to the hospital.

### Endpoint

The endpoint was hospitalization with clinical deterioration as described above the emergency protocol. In addition, the entire population was divided into two groups, depending on whether they were hospitalized or discharged home.

### Ethical considerations

This study conformed to the principles of the Declaration of Helsinki and was approved by the Institutional Review Board of Mie University Graduate School of Medicine and each participating hospital ethics committee (Reference number H2021-185). All patients gave their "opt-out" informed consent for the study. Data were acquired from electronic medical records and anonymized after collection. Patients under the age of 15 years were excluded due to ethical considerations.

### Statistical analysis

Continuous variables with normal distributions are expressed as means ± standard deviation (SD) according to the distribution of the data. Non-normally distributed data are reported as medians [interquartile range (IQR)]. Categorical variables are expressed as percentages (%) unless otherwise indicated. Baseline characteristics were compared with the chi-squared test or Fisher's exact test for categorical variables, and with Student's or Welch's $t$-test after testing for a normal distribution of continuous variables. Statistical analyses for multiple groups were performed by one-way ANOVA, followed by Holm's post hoc test for multiple comparisons. The post hoc tests were performed only if a p value < 0.05 was achieved, and there was no significant variance inhomogeneity. Logistic regression analysis was also performed to investigate the independent predictors of hospitalization, and odds ratio (OR) of each clinical risk factor for hospitalization was calculated.

A simple risk scoring system was created using the significant hospitalization factors. Points were assigned to correlate the score with the ORs determined by the logistic regression model, and the hospitalization rate was stratified by the number of scale points. Internal validation of the original data was performed using the bootstrap method with 200 repetitions of sampling to evaluate whether the scoring system could accurately predict the risk of hospitalization. Significance was defined as a p value < 0.05, and all statistical analyses were performed using EZR (version 1.60), which is a graphical user interface for R (version 4.2.1).

## Results

### Outcome

The median duration until the admission to the designated hotel was 3 days [IQR 2–4 days] after being diagnosed with COVID-19 by the real-time polymerase chain reaction test. During the study period, 1,087 COVID-19 patients were admitted to the designated hotel, and the following subjects were excluded: patients under the age of 15 years (n = 33, 3.0%), those admitted from the hospital (n = 111, 10.2%), COVID-19-vaccinated patients (n = 4, 0.4%), and

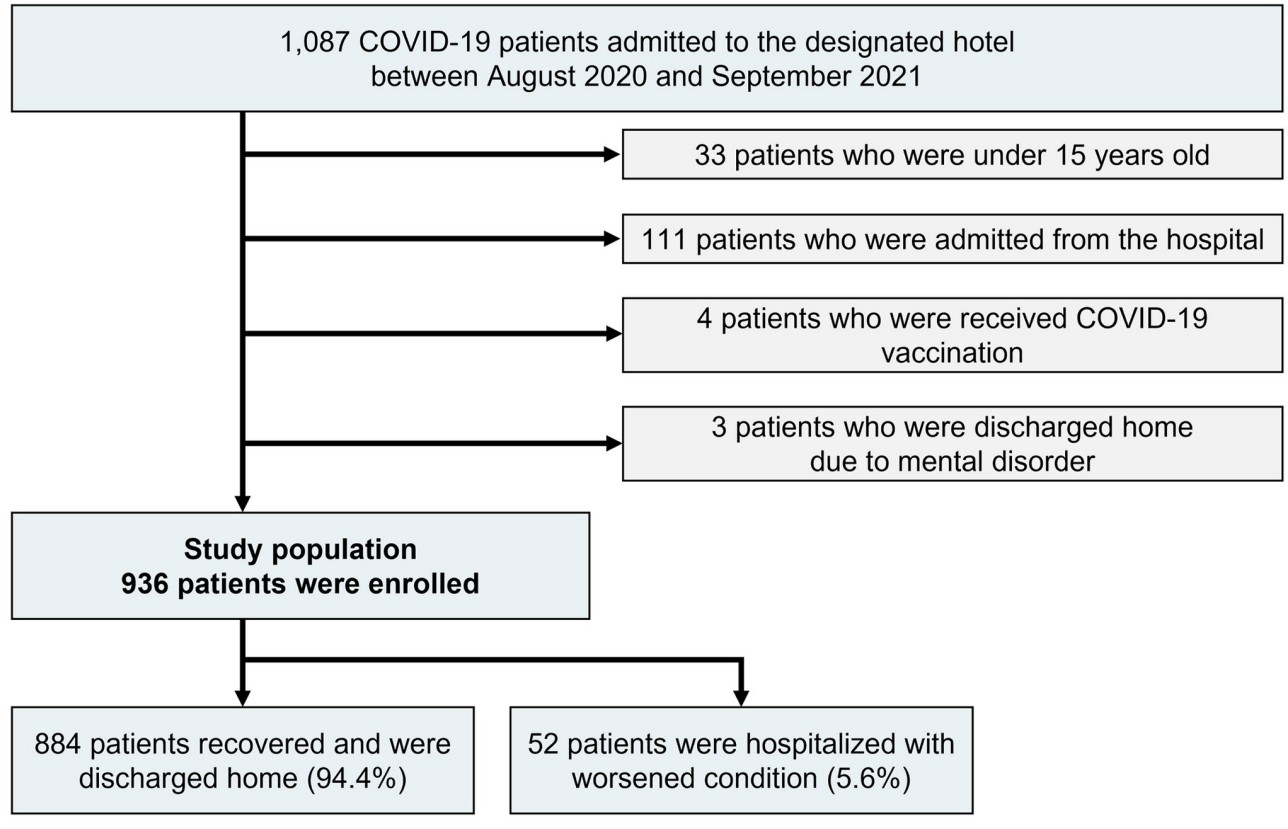

**Fig 1. Study population.**

patients who were discharged to their own home due to mental or social disorders (n = 3, 0.3%).

Finally, a total of 936 patients were included in the current analysis. Of them, 884 patients (94.4%) were discharged to their own home with improving symptoms, whereas 52 patients (5.6%) were hospitalized due to the deterioration of their conditions (Fig 1). However, no patients died or had serious complications from COVID-19 during the study period. The reasons for the hospitalization were persistent fever (N = 41, 78.8%), followed by poor oral intake (N = 19, 36.5%), oxygen desaturation (N = 17, 32.7%), and worsening respiratory symptoms (N = 11, 21.2%).

## Number of COVID-19 patients

Fig 2 shows the changes in the numbers of COVID-19 patients in Mie Prefecture from August 25, 2020 to September 31, 2021. The first patient was admitted to the designated hotel due to the increasing hospital bed occupancy rate in the 2nd wave. As the number of COVID-19 infections increased, the number of patients who were admitted to the designated hotel increased during the 3rd, 4th, and 5th waves. The hospital bed occupancy rate had also reached over 60% in early May 2021 due to increased social activities and spread of the Alpha variant (the 4th wave). Since August 2021, the bed occupancy rate reached about 70% in the 5th wave caused by the spread of the Delta variant, and the number of patients who were admitted to hotels or who recuperated at home increased. The number of infected people in this study according to the public health center is shown in S2 Table.

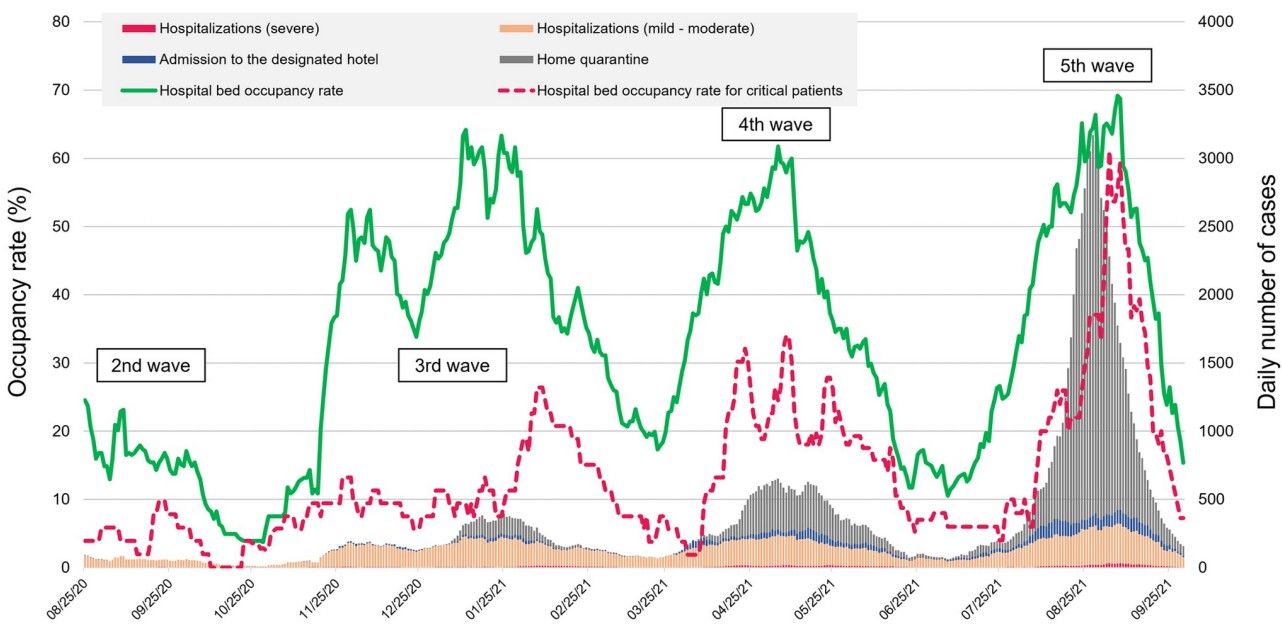

**Fig 2. Changes in the number of COVID-19-infected persons, those who were quarantined in their own home or admitted to the designated hotel, and the hospital bed occupancy rate in Mie Prefecture.**

## Patients' characteristics

Table 1 shows the patients' characteristics. The average age was 32.1 ± 12.8 years, and 61.3% were male. The mean age was higher (31.3 ± 12.4 vs 46.0 ± 11.9 years, p < 0.01), and BMI was higher (22.2 ± 3.2 vs 24.4 ± 3.3 kg/m$^2$, p < 0.01) in the hospitalized patients than in the patients who were discharged to their own homes. Hospitalized patients had a significantly higher prevalence of hypertension than those who were discharged to their own home. On hotel admission, the temperature was higher (36.7 ± 0.5 vs 37.4 ± 0.8˚C, p < 0.01) and the SpO$_2$ level was lower (98.3 ± 1.0 vs 97.4 ± 1.8%, p < 0.01) in the hospitalized patients compared with those who were discharged to their own home. Table 1 shows patients' symptoms during hotel accommodation. Of the total cases, 7.2% reported no symptoms during the clinical course. Compared with the patients who were discharged to their own home, the hospitalized patients had significant higher rates of fever (≥ 37.5˚C), cough, shortness of breath, fatigue, loss of appetite, nausea/vomiting, and diarrhea.

S3 Table presents the patients' characteristics divided into three groups according to the waves of COVID-19 infections. Due to the revision of the admission criteria, the age of patients of the 5th wave was higher, and they had a significantly higher prevalence of comorbidities (especially hypertension) compared to the those of the 2nd/3rd wave or 4th wave. Some symptoms, such as fever, cough, sore throat, and arthralgia/myalgia were more frequent in the patients of the 5th wave than of the 2nd/3rd wave or 4th wave, but there was no significant difference in the hospitalization rates.

## Risk of hospitalization

Table 2 reports the crude and adjusted ORs and 95% confidence intervals from the logistic regression analysis for hospitalization. The rates of hospitalization were not significantly changed among the waves of COVID-19 patient surge. The odds ratio increased with age and was significantly higher among patients aged > 50 years (OR = 14.2, 95% confidence interval:

**Table 1. Patients' characteristics and symptoms.**

| | All | Home discharge | Hospitalized | p value |
|---|---|---|---|---|
| No. of patients | 936 | 884 | 52 | |
| Age, years | 27 [21–42] | 26 [21–41] | 49 [40–55] | <0.01 |
| ≤29 years, % | 512 (54.7) | 504 (57.0) | 8 (15.4) | <0.01 |
| 30–39 years, % | 139 (14.9) | 134 (15.2) | 5 (9.6) | 0.37 |
| 40–49 years, % | 160 (17.1) | 144 (16.3) | 16 (30.8) | 0.01 |
| ≥50 years, % | 125 (13.4) | 102 (11.5) | 23 (44.2) | <0.01 |
| Male, % | 574 (61.3) | 538 (60.9) | 36 (69.2) | 0.29 |
| BW, kg | 62.8 ± 11.8[a] | 62.3 ± 11.8[b] | 68.9 ± 11.1[c] | <0.01 |
| BMI, kg/m$^2$ | 22.4 ± 3.3[d] | 22.2 ± 3.2[e] | 24.4 ± 3.3[c] | <0.01 |
| <18.5 kg/m$^2$, % | 73 (10.9)[d] | 71 (11.3)[e] | 2 (4.5)[c] | 0.25 |
| 18.5–25.0 kg/m$^2$, % | 457 (68.0)[d] | 434 (69.1)[e] | 23 (52.3)[c] | 0.03 |
| ≥25.0 kg/m$^2$ | 142 (21.1)[d] | 123 (19.6)[e] | 19 (43.2)[c] | <0.01 |
| Comorbidities | | | | |
| Hypertension | 43 (4.6) | 36 (4.1) | 7 (13.5) | <0.01 |
| Diabetes mellitus | 1 (0.1) | 1 (0.1) | 0 (0.0) | 1.00 |
| Dyslipidemia | 18 (1.9) | 16 (1.8) | 2 (3.8) | 0.26 |
| Hyperuricemia | 12 (1.3) | 10 (1.1) | 2 (3.8) | 0.14 |
| Malignancy (including history), % | 5 (0.5) | 5 (0.6) | 0 (0.0) | 1.00 |
| Asthma (including history), % | 17 (1.8) | 17 (1.9) | 0 (0.0) | 0.62 |
| Smoking, % | 207 (22.1) | 195 (22.1) | 12 (23.1) | 0.86 |
| Temperature on hotel admission, ˚C | 36.7 ± 0.5 | 36.7 ± 0.5 | 37.4 ± 0.8 | <0.01 |
| Temperature ≥37.0˚C at hotel admission, % | 290 (31.0) | 254 (28.7) | 36 (69.2) | <0.01 |
| SpO$_2$ on room air at hotel admission, % | 98.3 ± 1.1 | 98.3 ± 1.0 | 97.4 ± 1.8 | <0.01 |
| SpO$_2$ on room air at hotel admission ≥ 97%, % | 204 (21.8) | 181 (20.5) | 23 (44.2) | <0.01 |
| Symptoms during clinical course | | | | |
| Asymptomatic, % | 67 (7.2) | 67 (7.6) | 0 (0.0) | 0.05 |
| Fever (≥ 37.5˚C), % | 670 (71.6) | 619 (70.0) | 51 (98.1) | <0.01 |
| Fever (≥ 38.5˚C), % | 91 (9.8) | 57 (6.5) | 34 (66.7) | <0.01 |
| Max temperature, ˚C | 37.3 ± 0.7 | 37.2 ± 0.6 | 38.7 ± 0.8 | <0.01 |
| Cough, % | 490 (52.4) | 447 (50.6) | 43 (82.7) | <0.01 |
| Shortness of breath, % | 37 (4.0) | 18 (2.0) | 19 (36.5) | <0.01 |
| Sore throat, % | 438 (46.8) | 410 (46.4) | 28 (53.8) | 0.37 |
| Runny nose / nasal obstruction, % | 339 (36.2) | 322 (36.4) | 17 (32.7) | 0.69 |
| Olfactory dysfunction, % | 154 (16.5) | 151 (17.1) | 3 (5.8) | 0.05 |
| Dysgeusia, % | 151 (16.1) | 147 (16.6) | 4 (7.7) | 0.12 |
| Fatigue, % | 419 (44.8) | 371 (42.0) | 48 (92.3) | <0.01 |
| Loss of appetite, % | 35 (3.7) | 12 (1.4) | 23 (44.2) | <0.01 |
| Headache, % | 324 (34.6) | 308 (34.8) | 16 (30.8) | 0.65 |
| Nausea / Vomiting, % | 28 (3.0) | 19 (2.1) | 9 (17.3) | <0.01 |
| Diarrhea, % | 105 (11.2) | 92 (10.4) | 13 (25.0) | <0.01 |
| Arthralgia / Myalgia, % | 178 (19.0) | 163 (18.4) | 15 (28.8) | 0.09 |

BW: body weight; BMI: body mass index; SpO$_2$: percutaneous oxygen saturation.

[a]N = 645,

[b]N = 601,

[c]N = 44,

[d]N = 672,

[e]N = 628.

**Table 2. Predictors of hospitalization on logistic regression analysis.**

| | OR | 95% CI | p value |
|---|---|---|---|
| The waves of infection | | | |
| 2nd and 3rd (August 2020 to February 2021) | Ref | | |
| 4th (March 2021 to June 2021) | 0.78 | 0.30–2.02 | 0.614 |
| 5th (July 2021 to September 2021) | 1.36 | 0.55–3.38 | 0.504 |
| Age | | | |
| ≤29 | Ref | | |
| 30–39 | 2.35 | 0.76–7.30 | 0.139 |
| 40–49 | 7.00 | 2.94–16.7 | <0.001 |
| ≥50 | 14.2 | 6.18–32.6 | <0.001 |
| Male sex | 1.45 | 0.79–2.65 | 0.231 |
| BMI | | | |
| <18.5 | 0.53 | 0.12–2.30 | 0.398 |
| 18.5–25.0 | Ref | | |
| ≥25.0 | 5.48 | 1.24–24.2 | 0.025 |
| Hypertension | 3.66 | 1.55–8.69 | 0.003 |
| Dyslipidemia | 2.17 | 0.49–9.70 | 0.311 |
| Hyperuricemia | 3.50 | 0.75–16.4 | 0.112 |
| Smoking | 1.06 | 0.55–2.06 | 0.864 |

OR: Odds ratio; CI: confidence interval; BMI: body mass index.

6.18–32.6) than that in the other age groups. Presence of hypertension and obesity were also associated with hospitalization.

We created a simple scale based on the results of the logistic regression model, and classified patients based on their score on this scale as follows: Group 1 (score 0–1, 69.4%), Group 2 (score 2–3, 24.3%), and Group 3 (score 4–5, 6.3%) (Table 3). Fig 3 shows that the hospitalization rate according to the risk scale of COVID-19 severity identified in this study. As the new risk scale score increased, the hospitalization rate increased significantly: 2.0% at 0–1 points, 9.7% at 2–3 points, and 28.8% at 4–5 points (p < 0.001). The bootstrap method was used to internally validate the risk scoring, and the calibration curve was plotted after the original data were repeatedly sampled 200 times (Fig 4). The corrected C-index, calibration

**Table 3. Risk factors of hospitalization and adjustment factor points.**

| Risk factor | OR | Adjustment factor points |
|---|---|---|
| Age | | |
| 40–49 | 7.00 | **2** |
| ≥50 | 14.2 | **3** |
| BMI | | |
| ≥25.0 | 5.48 | **1** |
| Hypertension | 3.66 | **1** |
| **Risk group** | **Patients, n (%)** | **Risk score, points** |
| Group 1 | 650 (69.4) | 0–1 |
| Group 2 | 227 (24.3) | 2–3 |
| Group 3 | 59 (6.3) | 4–5 |

OR: Odds ratio; BMI: body mass index.

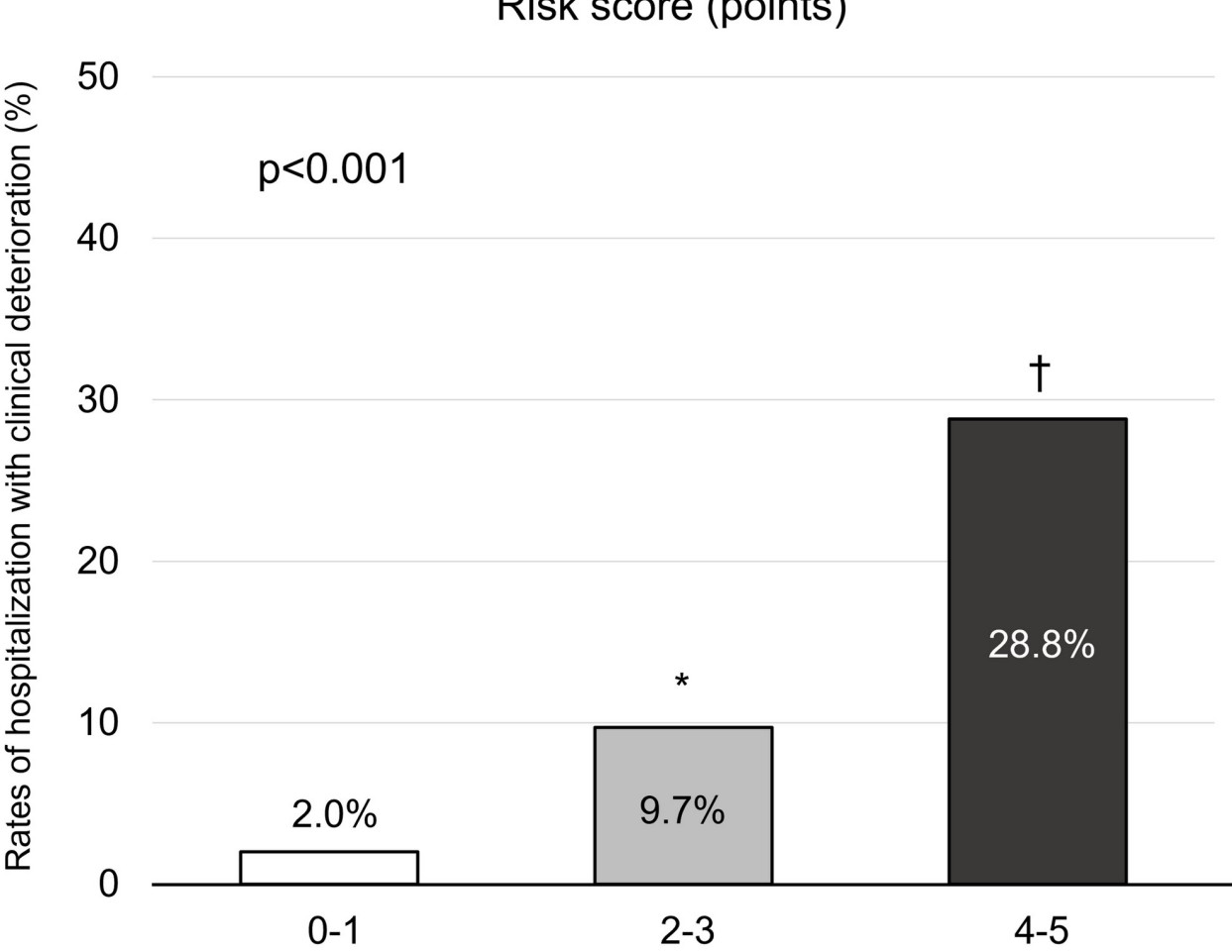

**Fig 3. The hospitalization rate according to the risk scale score of COVID-19 severity identified in this study.** The risk scale is shown in Table 3. Statistical analyses were performed with Fisher's exact test, followed by Holm's post hoc test for multiple comparisons. *p < 0.001 vs 0–1 points; † p < 0.001 vs 0–1 points and 2–3 points.

intercept, and calibration slope were 0.7612, 0.0412, and 1.0084, respectively. After bootstrap calibration, the mean absolute error was 0.009, and the Hosmer-Lemeshow goodness-of-fit test $\chi^2$ = 44.739, P = 0.5806, indicating that the model had good prediction accuracy.

## Discussion

The present study investigated the safety of the designated hotel accommodation program and identified risk factors for hospitalization in patients with COVID-19 in Japan. The major findings were: 1) around 6% of patients were hospitalized, but none of them died or developed serious complications from COVID-19; 2) older age, higher BMI, and presence of hypertension were the risk factors for hospitalization due to clinical deterioration; and 3) the new validated risk scale successfully identified high-risk patients for hospitalization. Furthermore, all patients in the present cohort had not yet received the first dose of COVID-19 vaccination. It is important to note that this study was based on data from the unvaccinated population. In Japan, the vaccination rate was low until June 2021, and it dramatically increased to about 50%

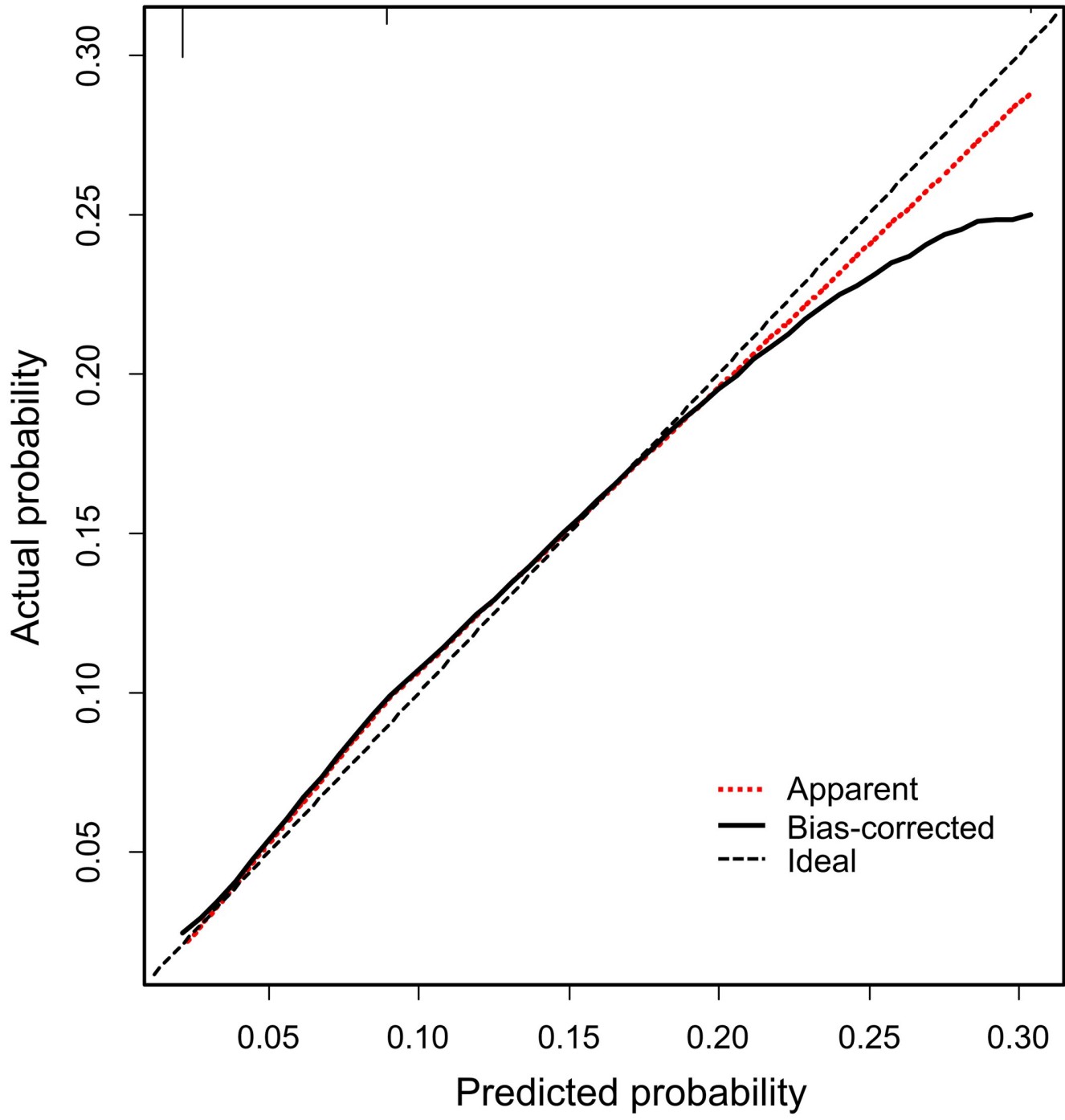

**Fig 4. Calibration curve with bootstrap resampling validation for predicting the risk of hospitalization.**

by September 2021. The number of vaccinated patients who were admitted to a designated hotel between August 2020 and September 2021 was very small in the present cohort, presumably because of the low vaccination rate and the high effectiveness of vaccination in the younger population.

## Role and purpose of admission to the hotel

Many countries may not have enough hospital beds for isolating and treating COVID-19 patients [8, 9]. For example, Wuhan in China faced a lack of critical care resources [10]. In Singapore, Hong Kong, and the state of Wisconsin in the USA, community isolation facilities for COVID-19 patients were operated, and the admission of COVID-19 patients to the isolation facility in Singapore was effective in keeping bed occupancy rates low [11].

In Japan, all cases infected with COVID-19 were required to be quarantined with hospitalization based on the initial law. However, a previous study indicated that the Japanese healthcare system and disaster response capabilities had not functioned well in the face of the COVID-19 pandemic due to its dispersed medical resources [12]. The Mie Prefecture Government requested each medical institution to secure the beds for all patients with COVID-19. In addition, 240 rooms for recuperating patients were prepared in the designated hotel in August 2020, and another facility was assigned a designated hotel in preparing for the spread of infection in advance in June 2021. Since the 4th wave, the coordinating team of the prefectural office and public health center aimed to decrease the hospital bed occupancy rate in the face of a further increase in COVID-19 patients, and for that purpose, they endorsed home-isolation for low-risk individuals with no or mild symptoms. In retrospect, the system of admission to the designated hotel combined with home-isolation contributed to saving the limited medical resources and hospital beds.

## Safety of recovering at the hotel

Recently, Sakamoto et al. has reported the clinical characteristics of COVID-19 patients who were admitted to public accommodation facilities and risk factors for hospitalization in metropolitan Tokyo [13]. In their research, 6.4% of patients were hospitalized, comparable to the present result (5.9%). Although the accommodation program was managed by each local government, the hospitalization rates were similar between a relatively sparsely populated rural prefectures and a densely populated metropolitan Tokyo areas. Furthermore, no fatal case was observed in the designated accommodation program during hotel stays in Mie Prefecture. However, there is some concern that sufficient support may not be provided depending on the number of infected patients or the scale of the designated accommodation. For example, a man in his 50s with COVID-19 and a history of heart disease was admitted to the designated hotel in Kanagawa Prefecture and died in August 2021 [14].

In each facility in Japan, revision of hotel admission criteria was required according to the COVID-19 infection status. As the hospital bed occupancy rate increases, higher risk patients should be accepted for hotel admission. Indeed, the upper limit age for acceptance was raised to 50 years in the 4th wave and 65 years in the 5th wave, and the admission criteria related to the clinical condition including patient symptoms and comorbidities for acceptance were also relaxed in the 4th and 5th waves in Mie Prefecture. Medical staff stationed at the hotel usually needed to assess the condition or severity of admitted patients from the vital data and symptoms recorded by the patients themselves. Also, in principle, there is no monitoring system and no routine visit from the perspective of infection risk. Thus, it is difficult to manage a sudden change in the patient's condition immediately. Considering this fact, appropriate risk management for COVID-19 severity on hotel admission is very important for safe recovery at the hotel.

## Risk management of COVID-19 severity and hotel admission

There are many reports of the risk of worsening severity of COVID-19. A previous meta-analysis reported that the proportion of diabetes mellitus, cardiovascular disease, and respiratory

disease was significantly higher in the critically or mortally ill group compared with the non-critically ill group. Clinical manifestations such as fever, shortness of breath, or dyspnea could also imply the progression of COVID-19 [15]. A cohort study in New York demonstrated that severe obesity (BMI $\geq$ 35 kg/m$^2$), older age (> 50 years), and male sex were independently associated with higher in-hospital mortality [16]. Public authorities including the Robert Koch Institute and the U.S. CDC reported that older age is a risk factor for severe and fatal disease courses of COVID-19, and some publications identified the age for increased risk to be from 50–60 years [17] and 65 years [18, 19]. Another previous study reported that the maximum temperature during COVID-19 infection was significantly correlated with the mortality rate, and there was a significant increase in mortality for every 0.5˚C increase from 37.0˚C [20]. An elevated temperature may indicate COVID-19 activity and may increase metabolic demand and oxygen consumption of multiple organs.

Obesity is also one of the most important risk factors for severe COVID-19. The U.S. CDC reported that the adjusted risk ratio for hospitalization was increased in patients with BMI $\geq$ 30 kg/m$^2$ compared with those with a BMI of 18.5–24.9 kg/m$^2$ among patients in whom approximately 79% were overweight or obese [21]. SpO$_2$ is also useful in COVID-19 patients for monitoring acute respiratory failure [22]. A previous study reported that patients with an SpO$_2$ of less than 92% at rest were more likely to need hospitalization than were patients with higher SpO$_2$ levels [23].

Furthermore, Ninomiya et al. in Japan showed that overweight (BMI $\geq$ 25 kg/m$^2$) was one of the independent risk factors for oxygen administration because of pneumonia or a low SpO$_2$ level ($\leq$ 93%) at rest among hospitalized COVID-19 patients with mild symptoms on admission [24]. Many previous reports suggested that patients with obesity had a background of hypoxemia or respiratory failure associated with pulmonary restriction, ventilation-perfusion mismatch, respiratory muscle fatigue, and sleep apnea syndrome [25, 26]. Therefore, one reason for the low COVID-19 mortality rate in Japan may be that the rate of obese people is lower than in Western countries. Thus, the optimal cut-off for BMI may be associated with race or ethnicity. In fact, only 21% of patients had BMI $\geq$ 25 kg/m$^2$ in the present study. Sakamoto et al. reported that age over 40 years and BMI $\geq$ 25 kg/m$^2$ were the risks for hospitalization in metropolitan Tokyo [13]. Importantly, the present study presented a novel risk score to identify the high-risk patients based on simple parameters including age, BMI, body temperature and SpO$_2$. Since body temperature and SpO$_2$ are essential elements in the management of respiratory infections, this novel risk scoring system can be a reliable method to identify COVID-19 patients at high risk of hospitalization.

On the other hand, a recent randomized, controlled trial reported that monitoring of SpO$_2$ in COVID-19 patients at home did not predict clinical deterioration. There was no significant difference in the number of days alive without hospitalization at 30 days between the pulse oximetry group and the standard program group. In that study, care escalation was triggered if the SpO$_2$ was $\geq$ 3% lower than the baseline first SpO$_2$ measurement (95–100%) [27]. Therefore, the present cut-off value of SpO$_2$, which is a higher threshold than previously reported, may be useful for the safe management of patients admitted to the designated hotel according to the criteria.

The present findings may contribute to constructing an optimal risk stratification system for safe management in the outpatient accommodation program for patients with COVID-19. Since the majority of patients in the present study were previously healthy and young members of the Japanese population, the optimal threshold of each risk was relatively milder than in previous reports. The independent risk factors for clinical deterioration and their optimal cut-off values may vary depending on patient background characteristics, including race, ethnicity, age range, and comorbidities.

## Limitations

This study has several potential limitations. First, because this was an observational, single-center study, unexpected and unmeasured confounders may have affected the outcomes. Second, the decisions to transfer patients to the hospital were at the discretion of the attending physicians or nurses. Therefore, the potential for selection bias cannot be excluded. A well-designed, multi-center, large-scale, prospective trial is needed to confirm the present findings. Third, the present findings may not be generally applied to populations outside of the hotel accommodation program, such as patients advised to stay at home. Fourth, surveillance tests to identify SARS-CoV-2 variants were not performed in all cases, though the characteristics and the variants of SARS-CoV-2 differed. Fifth, the data of home isolation were not available, and it was not possible to evaluate the differences in safety between hotel accommodation and home isolation. If families or roommates monitored the patients isolated at home, home quarantine may be safer than hotel isolation. Finally, although the operation and risk management of the hotel accommodation program were found to be safe in this study, whether this program was effective for reducing the mortality or other serious complication of COVID-19 remains unclear.

## Conclusions

The designated hotel program for COVID-19 patients was safe, especially for those with a low risk for hospitalization in Japan. Age $\geq$ 40 years, BMI $\geq$ 25 kg/m$^2$, and presence of hypertension were the risk factors for hospitalization, and the new risk score successfully identified the higher risk patients.

## Supporting information

**S1 Fig. Flow chart of COVID-19 patients.**
(TIF)

**S1 Table. Admission criteria and the waves of infection.**
(DOCX)

**S2 Table. Number of infected people reported by each public health center.**
(DOCX)

**S3 Table. Patients' characteristics stratified by the period of admission to the hotel.**
(DOCX)

## Acknowledgments

The authors are grateful to the coordinating team of the Department of Medical Health, Mie Prefectural Government and all staff working at the hotel.

## Author Contributions

**Formal analysis:** Hiromasa Ito.

**Investigation:** Hiromasa Ito.

**Project administration:** Hiromasa Ito, Tairo Kurita, Masahiro Hirayama, Shuji Isaji.

**Supervision:** Shuji Isaji, Kaoru Dohi.

**Validation:** Tadafumi Sugimoto, Yoshito Ogihara, Masaki Tanabe, Masahiro Hirayama.

**Writing – original draft:** Hiromasa Ito.

**Writing – review & editing:** Yoshito Ogihara, Kaoru Dohi.

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
