## [Decision Letter · Decision Letter 0]

21 Sep 2022

PONE-D-22-22615Clinical Characteristics and the Risk of Hospitalization of Patients with Coronavirus Disease 2019 Quarantined in a Designated Hotel in JapanPLOS ONE

Dear Dr. Dohi,

Thank you for submitting your manuscript to PLOS ONE. After careful consideration, we feel that it has merit but does not fully meet PLOS ONE’s publication criteria as it currently stands. Therefore, we invite you to submit a revised version of the manuscript that addresses the points raised during the review process.

The manuscript should be revised according to the Reviewer's suggestions. AE did not see the second reviewer for this manuscript; however, AE decided to avoid the delay in the fate of this manuscript. See the Reviewer's comments carefully and respond to them appropriately.

We look forward to receiving your revised manuscript.

Kind regards,

Masaki Mogi

Academic Editor

PLOS ONE

Journal Requirements:

2. In ethics statement in the manuscript and in the online submission form, please provide additional information about the patient records/samples used in your retrospective study. Specifically, please ensure that you have discussed whether all data/samples were fully anonymized before you accessed them and/or whether the IRB or ethics committee waived the requirement for informed consent. If patients provided informed written consent to have data/samples from their medical records used in research, please include this information.

4. We note that Figure S3 in your submission contain [map/satellite] images which may be copyrighted. All PLOS content is published under the Creative Commons Attribution License (CC BY 4.0), which means that the manuscript, images, and Supporting Information files will be freely available online, and any third party is permitted to access, download, copy, distribute, and use these materials in any way, even commercially, with proper attribution. For these reasons, we cannot publish previously copyrighted maps or satellite images created using proprietary data, such as Google software (Google Maps, Street View, and Earth). For more information, see our copyright guidelines: http://journals.plos.org/plosone/s/licenses-and-copyright.

a. You may seek permission from the original copyright holder of Figure S3 to publish the content specifically under the CC BY 4.0 license.  

Reviewers' comments:

Reviewer's Responses to Questions

**Comments to the Author**

1. Is the manuscript technically sound, and do the data support the conclusions?

Reviewer #1: Partly

2. Has the statistical analysis been performed appropriately and rigorously? 

Reviewer #1: No

3. Have the authors made all data underlying the findings in their manuscript fully available?

Reviewer #1: No

4. Is the manuscript presented in an intelligible fashion and written in standard English?

Reviewer #1: Yes

5. Review Comments to the Author

Reviewer #1: This study was a single-center, retrospective, observational study conducted at a COVID-19-specific quarantine facility in Japan.

The study was mainly descriptive; however, the authors have also developed a predictive model to predict hospitalization from the quarantine facility.

The following points need to be revised appropriately for publication.

<major concerns="">

(1) I'm afraid the study concerning an isolation facility in Tokyo, Japan, has already been published. (Sakamoto, et al. PMID: 35717438) Since the sample size of this study was smaller than that of the previous research, it is necessary to show the strengths of this study compared to the previous research. The authors need to focus on the novelty of this study.

(2) The prediction model's method is not clearly stated. In other words, the method of making the predictive model seemed somewhat arbitrary: one of my concerns is the cutoff value (e.g., age of 37 and BMI of 24), and the other concern is the fitting of the model.

How did you decide on the fitting of the model? Did you collect only those that were clinically significant or only those that were statistically significant? Further mention would be needed on this point. If this point cannot be addressed, the study should be revised, focusing on a descriptive study.

(3) Since the model did not do the external validation, the model had severe limitations. Could the authors perform external validations using another dataset or boot-strap method?

<minor concerns="">

(1) Regarding Table S1, did the all patients take blood or radiographic tests before the hotel admission?

(2)Were the patients with mental disorders included in the hotel-quarantine patients? If so, could that information (having mental illnesses or not) be added to the covariates?

(3) As for the criteria of discharge from the quarantine hotel, did it need "COMPLETE" improvement of the patient's symptoms? This seems to differ from the criteria generally used in Japan. In Japan, the end of isolation requires that “respiratory symptoms are improving.” (https://www.mhlw.go.jp/stf/covid-19/qa.html)

(4) Although the authors said "does not provide medical treatment (line 75)", the authors also described "the doctor could also prescribe oral medications for the patients if necessary. (line 117)". This seems to be confusing for readers outside Japan. The authors should state that "face-to-face medical care, including oxygen administration and intravenous fluid therapy, could not be provided" and "however, telemedicine, including drug prescription (without anti-viral drugs), could be provided."

(5) Were all patients with fever lasting more than 2 (consecutive?) days transported and admitted to the hospital? Or did medical providers "consider" transportation? This difference may be significant because of selection bias. The authors should describe it in detail.

(6) was the age of patients normally distributed? If not, the authors should change the notation using median and IQR.

(7) Although about half of the Japanese people had been vaccinated as of September 2021, the number of vaccinated patients in this cohort was too small. Could the authors explain the reason?

(8) The authors should include the waves of COVID-19 surge in the covariate. This is because the characteristics and pathogen of the variants of SARS-CoV-2 (e.g., alpha variant, delta variant…) differed. For the authors, it may be challenging to identify the SARS-CoV-2 variant of the eligible patients. The waves of COVID-19 should be a surrogate covariate for the SARS-CoV-2 variants.

(9) Could the authors show the reference of the sentence in Lines 279-281 in terms of privacy protection? The readers outside Japan may wonder how the authors knew this information.

(10) The authors should discuss the difference between isolation in hotels and isolation at home. For example, their families monitored the patients staying in their homes. While the patients isolated in the hotels might die without anyone noticing because the room in the hotel does not have a security camera and because the patients communicate with someone only by the telephone or some SNS tool. This may imply home quarantine may be safer than hotel isolation. Could the authors show the merit of hotel isolation using the data in this study?

(11) In Line 335, the sentence, "The findings might suggest that the timing of care escalation was too late," would be an intuitive leap. If the authors' opinion was correct, an overwhelming number of patients must have been transported to the hospitals, and as a result, the healthcare system would collapse.</minor></major>

6. PLOS authors have the option to publish the peer review history of their article (what does this mean?). If published, this will include your full peer review and any attached files.

Reviewer #1: No

---

## [Author Response · Author response to Decision Letter 0]

11 Nov 2022

Response to reviewer:

Thank you very much for your suggestions. Your comments were very useful for improving our manuscript. According to your and the other reviewers’ comments, we have updated our manuscript, Table 1, Table 2, Figure 4, and S3 Table. 

Major Comments

1. I'm afraid the study concerning an isolation facility in Tokyo, Japan, has already been published. (Sakamoto, et al. PMID: 35717438) Since the sample size of this study was smaller than that of the previous research, it is necessary to show the strengths of this study compared to the previous research. The authors need to focus on the novelty of this study.

RESPONSE:

Thank you for your important suggestion. As you pointed out, the results from Sakamoto et al. were similar to those of our research. However, 1) this is the first report to investigate the clinical characteristics of COVID-19 patients who were admitted to a designated hotel in rural a prefecture, and 2) to present the novel risk score to identify the high-risk patients based on simple parameters including age, BMI, body temperature and SpO2.

We added the reference you provided, and revised the sentence in the introduction and discussion section as follows:

[Introduction]

“Therefore, the purpose of the present study was to investigate the clinical characteristics of COVID-19 patients who were admitted to a designated hotel, and to clarify the safety of the designated hotel accommodation program and identify the risk factors for hospitalization during outbreaks or pandemic periods using the medical records in Mie Prefecture, Japan.” (Page 3, Line 57 – Page 4, Line 60)

[Discussion] 

“The present study is the first to have investigated the safety of the designated hotel accommodation program and identified risk factors for hospitalization in patients with COVID-19 in Japan.” (Page 18, Line 268–270)

“… and 3) the new validated 7-point risk scale successfully identified high-risk patients for hospitalization.” (Page 18, Line 273)

“Recently, Sakamoto et al. has reported the clinical characteristics of COVID-19 patients who were admitted to public accommodation facilities and risk factors for hospitalization in metropolitan Tokyo [Y Sakamoto, et al. J Infect Chemother. 2022;28:1439-1444.]. In their research, 6.4% of patients were hospitalized, comparable to the present result (5.9%). Although the accommodation program was managed by each local government, the hospitalization rates were similar between rural prefectures and Tokyo metropolitan areas. Furthermore, no fatal case was observed in the designated accommodation program during hotel stays in Mie Prefecture.” (Page 20, Line 301-307)

“Sakamoto et al. reported that age over 40 years and BMI ≥ 25 kg/m2 were the risks for hospitalization in metropolitan Tokyo [Y Sakamoto, et al. J Infect Chemother. 2022;28:1439-1444.]. Importantly, the present study presented a novel risk score to identify the high-risk patients based on simple parameters including age, BMI, body temperature and SpO2. Since body temperature and SpO2 are essential elements in the management of respiratory infections, this novel risk scoring system can be a reliable method to identify COVID-19 patients at high risk of hospitalization.” (Page 23, Line 359-364)

2. The prediction model's method is not clearly stated. In other words, the method of making the predictive model seemed somewhat arbitrary: one of my concerns is the cutoff value (e.g., age of 37 and BMI of 24), and the other concern is the fitting of the model.

How did you decide on the fitting of the model? Did you collect only those that were clinically significant or only those that were statistically significant? Further mention would be needed on this point. If this point cannot be addressed, the study should be revised, focusing on a descriptive study.

RESPONSE

Thank you for your important suggestion. We agree that the method in the original manuscript was poorly documented. At first, we identified the independent predictors of hospitalization using a logistic regression analysis by adjusting for clinically relevant factors, such as general background characteristics, comorbidities, and vital sign data with p values < 0.01 on univariate analysis (Please see New Table 2). Second, the optimal cut-off values of age, BMI, temperature, and SpO2 were investigated using ROC curve analyses (Please see S2 Figure). Third, we created new risk scale based on the results of adjusted odds ratios of binary logistic regression analysis.

We added New Table 2, revised Table 3 and revised the sentence in the method section as follows. 

“Logistic regression analysis was also performed to investigate the independent predictors of hospitalization, which was constructed by adjusting for clinically relevant factors, such as general background characteristics, comorbidities, and vital sign data with p values < 0.01 on univariate analysis. Then, the optimal cut-off values of the identified independent predictors of hospitalization such as age, BMI, temperature, and SpO2 were investigated using ROC curve analyses (S2 Figure). The odds ratio (OR) of each clinical risk factor for hospitalization was calculated by binary logistic regression analysis adjusted by age, male sex, BMI, hypertension, temperature at hotel admission, and SpO2 at hotel admission. A new risk scale based on the results of adjusted odds ratios was created.” (Page 9, Line 160-168)

“Table 3 shows that older age (≥ 37 years), higher BMI (≥ 24 kg/m2), higher temperature (≥ 37.0 °C), and lower SpO2 (≤ 97%) on hotel admission were the independent prognostic factors for hospitalization.” (Page 12, Line 223-225)

[Table legend]

Table 2. Predictors of hospitalization on logistic regression analysis

3. Since the model did not do the external validation, the model had severe limitations. Could the authors perform external validations using another dataset or boot-strap method?

RESPONSE

We appreciate the reviewer’s comment of this point. According to your advice, we performed internal validation using Boot-strap method. Our scoring model also had good discrimination (C-index, 0.8561) and calibration (Hosmer–Lemeshow test χ2 = 40.500, P value = 0.1958). These statistical analyses were performed using R (version 4.2.1).

We added Figure 4 and revised the sentence in the method and results section as follows. 

[Methods]

“Internal validation of the original data was performed using the bootstrap method with 200 repetitions of sampling to evaluate whether the scoring system could accurately predict the risk of hospitalization. Significance was defined as a p value < 0.05, and all statistical analyses were performed using EZR (version 1.60), which is a graphical user interface for R (version 4.2.1).” (Page 9, Line 171 - Page 10, Line 174)

[Results]

“The bootstrap method was used to internally validate the risk scoring, and the calibration curve was plotted after the original data were repeatedly sampled 200 times (Fig 4). The corrected C-index, calibration intercept, and calibration slope were 0.8561, 0.0294, and 1.0042, respectively. After bootstrap calibration, the mean absolute error was 0.009, and the Hosmer-Lemeshow goodness-of-fit test χ2 = 40.500, P = 0.1958, indicating that the model had good prediction accuracy.” (Page 17, Line 245-249)

[Figure legend]

Fig 4. Calibration curve with bootstrap resampling validation for predicting the risk of hospitalization.

Mainor Comments

1. Regarding Table S1, did the all patients take blood or radiographic tests before the hotel admission?

RESPONSE

Thank you for your important suggestion. 

“Hotel admission was permitted if the physician decided that the blood and radiographic examinations were not needed. Indeed, almost all patients (96.5%) who were admitted from their home did not undergo blood or radiographic examinations before the hotel admission.”

We added the above sentences in the section of the method. (Page 5, Line 87-90)

2. Were the patients with mental disorders included in the hotel-quarantine patients? If so, could that information (having mental illnesses or not) be added to the covariates?

RESPONSE

Thank you for your important suggestion. Unfortunately, there were no detailed data on mental disorders for each patient before hotel admission in the present study. 

3. As for the criteria of discharge from the quarantine hotel, did it need "COMPLETE" improvement of the patient's symptoms? This seems to differ from the criteria generally used in Japan. In Japan, the end of isolation requires that “respiratory symptoms are improving.” (https://www.mhlw.go.jp/stf/covid-19/qa.html) 

RESPONSE

We apologize for confused you. We revised the incorrect description as follows.

“The patients could be discharged from the hotel 10 days after the onset of symptoms and 48 hours after the resolution of fever or respiratory symptoms had improved.” (Page 7, Line 126-127)

4. Although the authors said "does not provide medical treatment (line 75)", the authors also described "the doctor could also prescribe oral medications for the patients if necessary. (line 117)". This seems to be confusing for readers outside Japan. The authors should state that "face-to-face medical care, including oxygen administration and intravenous fluid therapy, could not be provided" and "however, telemedicine, including drug prescription (without anti-viral drugs), could be provided."

REPONSE

Thank you for your valuable suggestions on our manuscript. We revised the manuscript according to your suggestion.

“Since the designated hotel is a non-medical facility and does not provide medical treatment, Doctors and nurses were dispatched from the cooperating medical institutions, including Mie University Hospital, and they were stationed at the hotel to monitor patients’ conditions and to respond to sudden deterioration. If the stationed doctors or nurses determined that a patient’s symptoms or condition had worsened, the coordinating team of the prefectural office quickly arranged for hospital treatment (S1 Figure). ” (Page 4, Line 74 – Page 5, Line 79)

“The designated hotel is a non-medical facility, and, therefore, face-to-face medical care including oxygen administration and intravenous fluid therapy could not be provided. However, if a patient described symptoms or requested a medical consultation, the on-call doctor examined the patient by telephone or video. Thus, telemedicine, including drug prescription (without anti-viral drugs), could be provided.” (Page 7, Line 120-124)

5. Were all patients with fever lasting more than 2 (consecutive?) days transported and admitted to the hospital? Or did medical providers "consider" transportation? This difference may be significant because of selection bias. The authors should describe it in detail.

RESPONSE

We completely agree with your suggestion. In our research, all patients with persistent fever (≥ 38.5 ℃ more than consecutive 2 days) were transported and admitted to the hospital. Then, we revised the manuscript as follows.

“…, if the patient developed sudden onset dyspnea, oxygen desaturation (SpO2 < 93%), chest pain, orthopnea, wheezing, difficulty breathing, ill complexion, persistent fever (≥ 38.5 ℃ more than 2 consecutive days), inadequate oral intake (less than 30% for more than 2 days), difficulty walking, or wound/fractures that required treatment. All patients who met the above criteria were transferred to the hospital.” (Page 7, Line 136-138)

6. Was the age of patients normally distributed? If not, the authors should change the notation using median and IQR.

RESPONSE

Thank you for your suggestion. As you pointed out, the age is non-normally distributed data. We revised Table 1, New S4 Table, and the manuscript as follows.

“Non-normally distributed data are reported as medians [interquartile range (IQR)].” (Page 9, Line 154-155)

7. Although about half of the Japanese people had been vaccinated as of September 2021, the number of vaccinated patients in this cohort was too small. Could the authors explain the reason?

RESPONSE

Thank you for your important suggestion. Until June 2021, the number of vaccinated people was low. Indeed, Sakamoto, et al. reported that the vaccinated patients who were admitted to the recovery accommodation facilities was also very low [Y Sakamoto, et al. J Infect Chemother. 2022;28:1439-1444.] (no patients were vaccinated between November 2020 and March 2021, 0.1% were vaccinated between April 2021 and May 2021, and 11.8% were vaccinated between June 2021 and July 2021. However, as you pointed out, the vaccinated rate in our study cohort was lower than expected. We revised the manuscript on this point as follows.

“In Japan, the vaccination rate was low until June 2021, and it dramatically increased to about 50% by September 2021. The number of vaccinated patients who were admitted to a designated hotel between August 2020 and September 2021 was very small in the present cohort, presumably because of the low vaccination rate and the high effectiveness of vaccination in the younger population.” (Page 18, Line 276-279)

8. The authors should include the waves of COVID-19 surge in the covariate. This is because the characteristics and pathogen of the variants of SARS-CoV-2 (e.g., alpha variant, delta variant…) differed. For the authors, it may be challenging to identify the SARS-CoV-2 variant of the eligible patients. The waves of COVID-19 should be a surrogate covariate for the SARS-CoV-2 variants.

RESPONSE

Thank you for your important suggestion. Unfortunately, surveillance tests to identify SARS-CoV-2 variants were not performed in all cases. Furthermore, as shown in S1 Table, the admission criteria were revised at every wave of COVID-19 infection in Mie prefecture. Based on the results, we did not use the waves of COVID-19 as a surrogate covariate for the SARS-CoV-2 variants. We added the below sentences in the limitations section.

“Fourth, surveillance tests to identify SARS-CoV-2 variants were not performed in all cases, though the characteristics and the variants of SARS-CoV-2 differed. Furthermore, the admission criteria were revised with every wave of COVID-19 infection in Mie prefecture. Thus, the waves of COVID-19 were not used as a surrogate covariate for the SARS-CoV-2 variants.” (Page 24, Line 387-391)

9. Could the authors show the reference of the sentence in Lines 279-281 in terms of privacy protection? The readers outside Japan may wonder how the authors knew this information.

RESPONSE

Thank you for your suggestion. According to your suggestion, we added the reference, Iwamoto S. Man with mild COVID-19 found in a state of cardiac arrest. The Asahi Shimbun. 2020 December 12 [Cited 2022 October 19]. Available from: https://www.asahi.com/ajw/articles/14013991.

Removed:

“Sudden deaths were also reported in Tokyo, Saitama, Osaka, and Hyogo Prefecture.” (Page 310, Line 310-311)

10. The authors should discuss the difference between isolation in hotels and isolation at home. For example, their families monitored the patients staying in their homes. While the patients isolated in the hotels might die without anyone noticing because the room in the hotel does not have a security camera and because the patients communicate with someone only by the telephone or some SNS tool. This may imply home quarantine may be safer than hotel isolation. Could the authors show the merit of hotel isolation using the data in this study?

RESPONSE

Thank you for the thoughtful comment. This point is very important, however, we don’t have the data of home isolation, and could not evaluate the differences between isolation in the hotel and isolation at home. So, we added the sentence to the limitation section as follows.

“Fifth, the data of home isolation were not available, and it was not possible to evaluate the differences in safety between hotel accommodation and home isolation. If families or roommates monitored the patients isolated at home, home quarantine may be safer than hotel isolation.” (Page 24, Line 391 – Page 25, Line 393)

11. In Line 335, the sentence, "The findings might suggest that the timing of care escalation was too late," would be an intuitive leap. If the authors' opinion was correct, an overwhelming number of patients must have been transported to the hospitals, and as a result, the healthcare system would collapse.

RESPONSE

We agree with your suggestion. According to your suggestion, we removed the sentence.

Response to Journal Requirements:

・We removed the map in S3 Figure, and changed the citation name from “S3 Figure” to “S2 Table”.

・We added the information about the patient records used in our study

“All patients gave their “opt-out” informed consent for the study. Data were acquired from electronic medical records and anonymized after collection.” (Page 8, Line 148-150)

・We added the Data Availability section and the sentence as follows.

 “Data cannot be shared publicly because of the Mie Prefectural Government policy. A request for deidentified data can be available from the corresponding author (dohik@med.mie-u.ac.jp).”

・We updated the URL of references due to the broken link (No. 6 and 17). 

・We revised the manuscript according to the editation by a native English speaker. (Page 2, Line 27, 28; Page 3, Line 48; Page 5, Line 91; Page 6, Line 98, 101; Page 10, Line 178-182; Page 11, Line 194; Page 24, Line 385; Page 25, Line 394-396, and the legends of Supplemental materials)

---

## [Decision Letter · Decision Letter 1]

5 Dec 2022

PONE-D-22-22615R1Clinical Characteristics and the Risk of Hospitalization of Patients with Coronavirus Disease 2019 Quarantined in a Designated Hotel in JapanPLOS ONE

Dear Dr. Dohi,

Thank you for submitting your manuscript to PLOS ONE. After careful consideration, we feel that it has merit but does not fully meet PLOS ONE’s publication criteria as it currently stands. Therefore, we invite you to submit a revised version of the manuscript that addresses the points raised during the review process.

The manuscript still needs major revisions.

We look forward to receiving your revised manuscript.

Kind regards,

Masaki Mogi

Academic Editor

PLOS ONE

Reviewers' comments:

Reviewer's Responses to Questions

**Comments to the Author**

1. If the authors have adequately addressed your comments raised in a previous round of review and you feel that this manuscript is now acceptable for publication, you may indicate that here to bypass the “Comments to the Author” section, enter your conflict of interest statement in the “Confidential to Editor” section, and submit your "Accept" recommendation.

Reviewer #1: (No Response)

2. Is the manuscript technically sound, and do the data support the conclusions?

Reviewer #1: Partly

3. Has the statistical analysis been performed appropriately and rigorously? 

Reviewer #1: No

4. Have the authors made all data underlying the findings in their manuscript fully available?

Reviewer #1: Yes

5. Is the manuscript presented in an intelligible fashion and written in standard English?

Reviewer #1: Yes

6. Review Comments to the Author

Reviewer #1: I thank the authors for the comments on my concerns.

While some points have been appropriately addressed, several problems remain unresolved, and I will provide some additional comments below.

#1

Initially, I thought that the novelty of this study was in the description of the patient characteristics who received treatments at public quarantine accommodations in Mie Prefecture. However, if the authors think that the main subject of this study is to assess the risk factors for hospitalization from the facilities, the authors and I need to discuss this further.

(1) Since patients are admitted to the hospital if their body temperature (BT) is over 38.5°C for more than two days, BT ≥37.0, one of the risk factors, seems to be an intermediate variable (not a confounding factor) of admission. In other words, it is obvious that patients with high body temperature will be transferred and admitted.

SpO2 may also be an intermediate factor; however, we can also interpret that SpO2 may be a confounding factor because hypoxia of hotel-admitted patients was derived from COPD or other lung diseases. The authors should reconsider whether these are the appropriate risk factors as well.

The authors should provide references that indicate that such intermediate variables are allowed to be included in the prediction model, or the authors should remove these vital signs from the prediction model.

(2) If the above risk factors are not included in the prediction model, the authors cannot show novelty compared with the previous studies.

This is a personal opinion, though; the fact that the population density is different from that of previous studies may lead to the novelty of this study.

Specifically, the novelty could be expressed by focusing on the difference between Tokyo, a densely populated area, and Mie, a relatively sparsely populated area.

However, this comment can be substantially different from the authors' thoughts, so if the authors have a good idea, please discard this opinion of mine.

(3) Can the age be categorized? If the authors want to create a scoring system, they should divide the population into 3-4 categories of age and score them according to odds ratios. This may show the difference from the previous studies.

#2

I could not understand why you did not include N waves of COVID-19 patient surge as a covariate.

If the admission criteria were changed depending on the waves, it is even more likely to be a confounding factor of the outcomes and should be included in the covariate.

I believe that regarding the waves of surge as covariates are not sufficient for addressing the confounding. If the authors consider that the heterogeneity of the patients in each wave is stronger, it is necessary to stratify the calculation of odds ratios by each wave.

At the very least, the fact that "different waves have different criteria for hospitalization" is not a reason to "not add waves to the covariate."

7. PLOS authors have the option to publish the peer review history of their article (what does this mean?). If published, this will include your full peer review and any attached files.

Reviewer #1: No

---

## [Author Response · Author response to Decision Letter 1]

20 Dec 2022

Response to reviewer:

Thank you very much for your suggestions. Your comments were very useful for improving our manuscript. According to your comments, we have updated our manuscript, Table 1, Table 2, Table 3, Figure 3, and Figure 4. 

1. Initially, I thought that the novelty of this study was in the description of the patient characteristics who received treatments at public quarantine accommodations in Mie Prefecture. However, if the authors think that the main subject of this study is to assess the risk factors for hospitalization from the facilities, the authors and I need to discuss this further.

(1) Since patients are admitted to the hospital if their body temperature (BT) is over 38.5°C for more than two days, BT ≥37.0, one of the risk factors, seems to be an intermediate variable (not a confounding factor) of admission. In other words, it is obvious that patients with high body temperature will be transferred and admitted.

SpO2 may also be an intermediate factor; however, we can also interpret that SpO2 may be a confounding factor because hypoxia of hotel-admitted patients was derived from COPD or other lung diseases. The authors should reconsider whether these are the appropriate risk factors as well.

The authors should provide references that indicate that such intermediate variables are allowed to be included in the prediction model, or the authors should remove these vital signs from the prediction model.

RESPONSE:

As you pointed out, body temperature and SpO2 seem to be an intermediate variable of hospitalization. We removed these variables from the regression analysis.

We revised Table 2, Table 3, Figure 3, Figure 4, and sentence as follows. 

[Abstract]

“The logistic regression analyses showed that older age (≥ 40 years), higher body mass index (≥ 25 kg/m2), and presence of hypertension were the independent risk factors for hospitalization. As the new risk scale score based on the results of the odds ratios increased, the hospitalization rate increased significantly: 2.0% at 0-1 points, 9.7% at 2-3 points, and 28.8% at 4-5 points (p < 0.001).” (Page 2, Line 30 – 35)

[Methods]

“Logistic regression analysis was also performed to investigate the independent predictors of hospitalization, and odds ratio (OR) of each clinical risk factor for hospitalization was calculated. ” (Page 9, Line 160 – 168)

“A simple risk scoring system was created using the significant hospitalization factors. Points were assigned to correlate the score with the ORs determined by the logistic regression model, and the hospitalization rate was stratified by the number of scale points.” (Page 9, Line 169 – Page 10, Line 173)

[Results]

“We created a simple scale based on the results of the logistic regression model, and classified patients based on their score on this scale as follows: Group 1 (score 0-1, 69.4%), Group 2 (score 2-3, 24.3%), and Group 3 (score 4-5, 6.3%) (Table 3). Fig 3 shows that the hospitalization rate according to the risk scale of COVID-19 severity identified in this study. As the new risk scale score increased, the hospitalization rate increased significantly: 2.0% at 0-1 points, 9.7% at 2-3 points, and 28.8% at 4-5 points (p < 0.001). The bootstrap method was used to internally validate the risk scoring, and the calibration curve was plotted after the original data were repeatedly sampled 200 times (Fig 4). The corrected C-index, calibration intercept, and calibration slope were 0.7612, 0.0412, and 1.0084, respectively. After bootstrap calibration, the mean absolute error was 0.009, and the Hosmer-Lemeshow goodness-of-fit test χ2 = 44.739, P = 0.5806, indicating that the model had good prediction accuracy.” (Page 13, Line 240 – Page 14, Line 250)

[Discussion]

“The major findings were: 1) around 6% of patients were hospitalized, but none of them died or developed serious complications from COVID-19; 2) older age, higher BMI, and presence of hypertension were the risk factors for hospitalization due to clinical deterioration; and 3) the new validated risk scale successfully identified high-risk patients for hospitalization.” (Page 19, Line 299 – 304)

Removed:

“The present study showed that older age, higher BMI, higher temperature, and lower SpO2 were the independent risk factors associated with deterioration and hospitalization. However, the cut-off values of any risks were milder in the current cohort population than in previous studies. These results may be mainly due to differences in race or ethnicity and the hotel admission criteria. At the start of hotel recuperation, the hotel admission criteria were set based on the following criteria: under 40 years of age, asymptomatic or mild/improving symptoms, no preexisting comorbidities, and BMI ≤ 30 kg/m2.” (Page 23, Line 375 – 381)

[Conclusions]

“Age ≥ 40 years, BMI ≥ 25 kg/m2, and presence of hypertension were the risk factors for hospitalization, and the new risk score successfully identified the higher risk patients.” (Page 26, Line 429 - 432)

(2) If the above risk factors are not included in the prediction model, the authors cannot show novelty compared with the previous studies.

This is a personal opinion, though; the fact that the population density is different from that of previous studies may lead to the novelty of this study.

Specifically, the novelty could be expressed by focusing on the difference between Tokyo, a densely populated area, and Mie, a relatively sparsely populated area.

However, this comment can be substantially different from the authors' thoughts, so if the authors have a good idea, please discard this opinion of mine.

RESPONSE:

Thank you for your valuable suggestions on our manuscript. We revised the manuscript in the discussion section according to your suggestion.

“Although the accommodation program was managed by each local government, the hospitalization rates were similar between a relatively sparsely populated rural prefectures and a densely populated metropolitan Tokyo areas.” (Page 21, Line 335 – 337)

(3) Can the age be categorized? If the authors want to create a scoring system, they should divide the population into 3-4 categories of age and score them according to odds ratios. This may show the difference from the previous studies.

RESPONSE:

Thank you for your important suggestion. We evaluated the risk of hospitalization by stratified age and BMI. Furthermore, we scored them according to the odds ratios.

We revised Table 1, Table 3 and the sentence as follows.

[Results]

“The odds ratio increased with age and was significantly higher among patients aged > 50 years (OR = 14.2, 95% confidence interval: 6.18-32.6) than that in the other age groups. Presence of hypertension and obesity were also associated with hospitalization.” (Page 13, Line 236 - 239)

[Figure legend]

“Fig 3. The hospitalization rate according to the risk scale score of COVID-19 severity identified in this study. The risk scale is shown in Table 3. Statistical analyses were performed with Fisher’s exact test, followed by Holm’s post hoc test for multiple comparisons. *p < 0.001 vs 0-1 points; † p < 0.001 vs 0-1 points and 2-3 points.”

2. I could not understand why you did not include N waves of COVID-19 patient surge as a covariate.

If the admission criteria were changed depending on the waves, it is even more likely to be a confounding factor of the outcomes and should be included in the covariate.

I believe that regarding the waves of surge as covariates are not sufficient for addressing the confounding. If the authors consider that the heterogeneity of the patients in each wave is stronger, it is necessary to stratify the calculation of odds ratios by each wave.

At the very least, the fact that "different waves have different criteria for hospitalization" is not a reason to "not add waves to the covariate."

RESPONSE

Thank you for your important suggestion. We agree with your opinion. We calculated the odds ratios using the logistic regression model.

We revised Table 2 and revised the sentence as follows. 

[Results]

“The rates of hospitalization were not significantly changed among the waves of COVID-19 patient surge.” (Page 13, Line 235 - 236)

[Limitations]

Removed:

“Furthermore, the admission criteria were revised with every wave of COVID-19 infection in Mie prefecture. Thus, the waves of COVID-19 were not used as a surrogate covariate for the SARS-CoV-2 variants.” (Page 25, Line 419 – Line 421)

We moved the following sentence to Page 12, Line 222.

“S3 Table presents the patients’ characteristics divided into three groups according to the waves of COVID-19 infections. Due to the revision of the admission criteria, the age of patients of the 5th wave was higher, and they had a significantly higher prevalence of comorbidities (especially hypertension) compared to the those of the 2nd/3rd wave or 4th wave. Some symptoms, such as fever, cough, sore throat, and arthralgia/myalgia were more frequent in the patients of the 5th wave than of the 2nd/3rd wave or 4th wave, but there was no significant difference in the hospitalization rates.”

---

## [Decision Letter · Decision Letter 2]

26 Dec 2022

Clinical Characteristics and the Risk of Hospitalization of Patients with Coronavirus Disease 2019 Quarantined in a Designated Hotel in Japan

PONE-D-22-22615R2

Dear Dr. Dohi,

We’re pleased to inform you that your manuscript has been judged scientifically suitable for publication and will be formally accepted for publication once it meets all outstanding technical requirements.

Kind regards,

Masaki Mogi

Academic Editor

PLOS ONE

Additional Editor Comments (optional):

Reviewers' comments:

Reviewer's Responses to Questions

**Comments to the Author**

1. If the authors have adequately addressed your comments raised in a previous round of review and you feel that this manuscript is now acceptable for publication, you may indicate that here to bypass the “Comments to the Author” section, enter your conflict of interest statement in the “Confidential to Editor” section, and submit your "Accept" recommendation.

Reviewer #1: All comments have been addressed

2. Is the manuscript technically sound, and do the data support the conclusions?

Reviewer #1: Yes

3. Has the statistical analysis been performed appropriately and rigorously? 

Reviewer #1: Yes

4. Have the authors made all data underlying the findings in their manuscript fully available?

Reviewer #1: Yes

5. Is the manuscript presented in an intelligible fashion and written in standard English?

Reviewer #1: Yes

6. Review Comments to the Author

Reviewer #1: I thank the authors for their great work.

I believe the authors have adequately addressed all issues.

7. PLOS authors have the option to publish the peer review history of their article (what does this mean?). If published, this will include your full peer review and any attached files.

Reviewer #1: No

---

## [Editor Report · Acceptance letter]

5 Jan 2023

PONE-D-22-22615R2 

Clinical Characteristics and the Risk of Hospitalization of Patients with Coronavirus Disease 2019 Quarantined in a Designated Hotel in Japan 

Dear Dr. Dohi:

I'm pleased to inform you that your manuscript has been deemed suitable for publication in PLOS ONE. Congratulations! Your manuscript is now with our production department. 

Kind regards, 

on behalf of

Dr. Masaki Mogi 

Academic Editor

PLOS ONE